# Catalase (CAT) Gene Family in Oil Palm (*Elaeis guineensis* Jacq.): Genome-Wide Identification, Analysis, and Expression Profile in Response to Abiotic Stress

**DOI:** 10.3390/ijms25031480

**Published:** 2024-01-25

**Authors:** Lixia Zhou, Jerome Jeyakumar John Martin, Rui Li, Xianhai Zeng, Qiufei Wu, Qihong Li, Dengqiang Fu, Xinyu Li, Xiaoyu Liu, Jianqiu Ye, Hongxing Cao

**Affiliations:** 1National Key Laboratory for Tropical Crop Breeding, Chinese Academy of Tropical Agricultural Sciences, Haikou 571101, China; lxzhou@catas.cn (L.Z.); jeromejeyakumarj@gmail.com (J.J.J.M.); irui@catas.cn (R.L.); zxh200888@126.com (X.Z.); qfi_wu@catas.cn (Q.W.); liqihong@catas.cn (Q.L.); fudq@catas.cn (D.F.); lixinyu@catas.cn (X.L.); liuxy86@catas.cn (X.L.); 2Coconut Research Institute, Chinese Academy of Tropical Agricultural Sciences, Wenchang 571339, China

**Keywords:** oil palm, CAT, genome-wide, abiotic stress, qPCR

## Abstract

Catalases (CATs) play crucial roles in scavenging H_2_O_2_ from reactive oxygen species, controlling the growth and development of plants. So far, genome-wide identification and characterization of CAT genes in oil palm have not been reported. In the present study, five *EgCAT* genes were obtained through a genome-wide identification approach. Phylogenetic analysis divided them into two subfamilies, with closer genes sharing similar structures. Gene structure and conserved motif analysis demonstrated the conserved nature of intron/exon organization and motifs among the *EgCAT* genes. Several cis-acting elements related to hormone, stress, and defense responses were identified in the promoter regions of *EgCAT*s. Tissue-specific expression of *EgCAT* genes in five different tissues of oil palm was also revealed by heatmap analysis using the available transcriptome data. Stress-responsive expression analysis showed that five *EgCAT* genes were significantly expressed under cold, drought, and salinity stress conditions. Collectively, this study provided valuable information on the oil palm CAT gene family and the validated *EgCAT* genes can be used as potential candidates for improving abiotic stress tolerance in oil palm and other related crops.

## 1. Introduction

Abiotic stress causes the accumulation of reactive oxygen species (ROS) in plant cells, and excessive ROS can lead to membrane lipid peroxidation and damage to the cell membrane [1]. Catalase (CAT) is the main antioxidant enzyme in the reactive oxygen species scavenging system produced by plants under abiotic stress. It can eliminate H_2_O_2_ produced in plants, alleviate the damage caused by oxidative stress to plants, and maintain cell stability [2,3]. CAT is a tetrameric heme protein composed of four subunits, mainly distributed in the peroxisome, glyoxylate cycle, and cytoplasm of plant cells, with a small amount distributed in mitochondria and chloroplasts [4]. CAT plays an important role in physiological processes, such as plant growth and development, stress response, defense response, and delaying plant age [5,6]. It mainly clears photorespiration, mitochondrial electron transfer, and β-hydrogen peroxide generated during fatty acid oxidation and other processes to prevent damage caused by reactive oxygen species to plants [7]. Enhancing CAT enzyme activity can reduce the accumulation of reactive oxygen species, thereby improving plant antioxidant capacity, which is of great significance for plant resistance to biotic and abiotic stresses. It provides a feasible approach for stress resistance breeding and improving crop resistance, and it has been widely applied in agricultural production [8]. Overexpression of CAT genes in crops like rice, corn, and cotton can mitigate the effects of abiotic stresses like cold, drought, and salinity on plants [9,10]. Members of the CAT family have been identified and analyzed in various crops, such as one CAT family member in castor bean (*Ricinus communis* L.) [11], tomato (*Solanum lycopersicum* L.) [12], and sweet potato (*Ipomoea batatas* L.) [13], three CAT members in tobacco (*Nicotiana plumbaginifolia*) [14] and rice (*Oryza sativa*) [15], and four *CAT* genes in cucumber (*Cucumis sativus* L.) [16]. So far, the rape (*Brassica napus* L.) has the most (ten) CAT family members [17].

CAT, a multi-gene family with diverse functions in plants, is influenced by both biotic and abiotic factors, exhibiting unique expression patterns over time. For instance, in sweet potato, SPCAT1 was an ethephon-inducible peroxisomal catalase, and its expression was regulated by reduced glutathione, DPI, EGTA, and cycloheximide. *SPCAT1* plays a physiological role in maintaining H_2_O_2_ homeostasis in leaves, influenced by developmental cues and environmental stimuli [13]. In another study, the overexpression of *Ibcat2* in *Escherichia coli* and Saccharomyces cerevisiae enhanced salt and drought tolerance, indicating its significant role in stress responses [18]. Overexpression of *ScCAT1* can enhance the biotic and abiotic stress of sugarcane, and this gene is involved in the protection of sugarcane against reactive oxidant-related environmental stimuli [19]. Mhamdi et al. [20] found that *AtCAT1* responded to abiotic stresses including cold, drought, and high salinity by clearing H_2_O_2._ However, *AtCAT2* and *AtCAT3* participated in the elimination of H_2_O_2_ subsidized by ROS homeostasis in light/dark conditions. Overexpression *AtCAT2* improved cold and drought resistance, while *AtCAT3* was predominantly stimulated by ABA and oxidative treatments [21]. Additionally, Contento and Bassham (2010) [22] observed chlorosis and necrotic scratches in *A. thaliana CAT2* mutants. The transcript level of *CAT1* in Ipomoea batatas was influenced by ethephon, condensed glutathione (GSH), nicotinamide adenine dinucleotide phosphate (NADPH), oxidase inhibitor diphenylene iodonium (DPI), calcium ion (Ca^2+^) chelator egtazic acid, and cycloheximide [13]. Furthermore, transgenic N. tabacum plants with the maize *CAT2* gene showed enhanced resistance to both oxidative stress and pathogen contagion [23].

In a similar way, the expression patterns of CAT genes vary in different tissues. For example, the expression of *NnCAT* genes in *Nelumbo nucifera* tissues was found to be highest in the young leaf and lowest in the roots. The mRNA level of *NnCAT* was significantly augmented in response to short-term mechanical wounding [24]. In cucumber, *CsCAT1* and *CsCAT2* showed a high expression in roots, leaves, and fruit, and *CsCAT3* was highly expressed in stems and leaves, while *CsCAT4* had no expression in any of these tissues. In addition, *CsCAT1*-*CsCAT3* were significantly induced under cold, drought, salinity, and ABA conditions [16]. In wheat, *TaCAT-A/B/D* were highly and exclusively expressed in all stages of leaf development; *TaCAT2* showed a low expression after anthesis; and *TaCAT3-A1/B/U* showed constitutive expression patterns. In addition, the expression of *TaCAT1* and *TaCAT2* decreased under fusarium treatment, while *TaCAT2* was induced using powdery mildew and septoria treatments [25].

In recent years, more and more studies have shown that under cold, drought, or salt stress, finding the candidate genes and dissecting signaling mechanisms in abiotic stress responses is a key determinant to developing abiotic stress-tolerant oil palm varieties. Li et al. (2019) investigated the expression levels of various genes (including *COR410*, *COR413*, *CBF1*, *CBF2*, *CBF3*, *ICE1-1*, *ICE1-2*, *ICE1-4*, *SIZ1-1*, *SIZ1-2*, *ZAT10*, *ZAT12*) and the accumulation of osmolytes in oil palm under cold stress and provided necessary information regarding the mechanism of the response and adaption of oil palm to cold stress [26]. However, there is currently a lack of research on the CAT gene family of oil palm. Oil palm (*Elaeis guineensis*) belongs to the palm family and is the most productive oil crop in the world [5,27]. However, when subjected to abiotic stresses, such as cold, drought, and high salt, palm oil production severely decreases. Therefore, a genome-wide comprehensive investigation and identification have been studied to obtain the CAT gene family in oil palm. Bioinformatics methods were used to predict and analyze their phylogenetic relationships, gene structures, evolutionary relationships, conserved motif domains, *cis*-acting elements, and tissue-specific expression. Fluorescence quantitative PCR was also used to analyze gene expression patterns under different stress conditions at different times and in different tissues. Altogether, the study provides a theoretical basis for in-depth research on the biological function of the *EgCAT* genes and the cultivation of high-quality and cold-resistant oil palm varieties.

## 2. Results

### 2.1. Identification of EgCAT Genes

A total of five CAT genes (named *EgCAT1*-*EgCAT5*) were identified from the oil palm genome via the bioinformatic method. Detailed information on these *EgCAT* genes is shown in Appendix A. Briefly, the length of genes ranges from 2158 bp (*EgCAT5*) to 6212 bp (*EgCAT2*). *EgCAT1*-*EgCAT4* comprises eight exons, while *EgCAT5* contains four exons. The coding DNA sequences (CDS) range from 2158 bp (*EgCAT5*) to 5925 bp (*EgCAT2*). The protein length of *EgCAT1*-*EgCAT4* is 492 amino acids, while the protein length of *EgCAT5* is 719 amino acids. The predicted molecular weights (MW) of *EgCAT1*-*EgCAT5* proteins ranged from 56.84 kDa (*EgCAT3*) to 78.95 kDa (*EgCAT5*). The isoelectric points (pI) of *EgCAT1*-*EgCAT5* were 6.94, 6.47, 6.99, 6.93, and 5.38, respectively. Analysis results of the subcellular localization showed that the EgCAT1 and EgCAT5 proteins were positioned on the cytoskeleton, the EgCAT2 and EgCAT3 proteins were positioned in the cytoplasm, and EgCAT4 was positioned on the chloroplast (Appendix A). In addition, 3, 3, and 11 CAT genes were identified from *Arabidopsis thaliana* (*AtCAT1*-*AtCAT3*), *Oryza sativa* (*OsCATA*, *OsCATB* and *OsCATC*), *Olea europaea* subsp. Europaea (*OeeCAT1*-*OeeCAT7*), and *Olea europaea* var. sylvestris (*OesCAT1*-*OesCAT4*) genome, respectively (Appendix A).

### 2.2. Phylogenetic Relationship and Structure Analysis of EgCAT Genes

A phylogenetic tree was constructed via the deduced protein sequences of the CAT genes of *Arabidopsis thaliana*, *Oryza sativa*, *Olea europaea* subsp. Europaea, and *Olea europaea* var. sylvestris. As shown in Figure 1, the obtained phylogenetic tree classifies all the genes into four groups. Group I contained nine CAT members (*OeeCAT1*-*2*, *OeeCAT4-7*, *AtCAT1-2,* and *OesCAT 4*). Group II was comprised of seven CAT genes (*OsCATA-C*, *EgCAT1-4*), which contained most of the CAT genes of oil palm. Group III was comprised of four CAT genes (*OeeCAT1-3* and *OesCAT3*). Group IV was comprised of two CAT genes (*AtCAT3* and *EgCAT5*). Our study revealed *EgCAT* genes’ closer phylogenetic relationship with *OsCATs.*

### 2.3. Chromosomal Distribution of EgCAT Genes

Three of the five *EgCAT* genes were located on the chromosomes of oil palm. *EgCAT1* was mapped on chromosome 4, *EgCAT2* was mapped on chromosome 12, and *EgCAT3* was mapped on chromosome 16 (Figure 2). *EgCAT4* and *EgCAT5* were distributed on two scaffold regions (p5_sc00387 and p5_sc33453) and have not yet been incorporated into the physical map of chromosomes (Appendix A).

### 2.4. Gene Structure and Conserved Motif Analysis of EgCAT Genes

The exon–intron number and distributions of all CAT genes from oil palm, *Arabidopsis thaliana*, *Oryza sativa*, *Olea europaea* subsp. Europaea, and *Olea europaea* var. sylvestris., were analyzed, and the structural information of CAT genes is shown in Figure 3. We found that most of the CAT genes contained 7–8 exons, only *EgCAT5* and *OsCATA* contained four exons, *OsCATC* contained six exons, and *OesCAT4* had nine exons.

Furthermore, the conserved motif composition was analyzed in each CAT gene from oil palm, *Arabidopsis thaliana*, *Oryza sativa*, *Olea europaea* subsp. Europaea, and *Olea europaea* var. sylvestris. via the online Multiple Expectation Maximizations for Motif Elicitation (MEME) tool (Figure 4). Motif analysis showed the occurrence of five conserved motifs (1–5) from the obtained CAT family members. The result indicated that all *AtCATs*, *EgCATs*, *OeeCATs*, *OesCATs,* and *OsCATs* contained 1–5 motifs except *EgCAT5.* Motifs 1, 3, 4, and 5 existed in *EgCAT5.* The amino acid composition of all motifs is illustrated in Figure 4.

### 2.5. Duplication Analysis of EgCAT Genes

MCScanx (https://github.com/wyp1125/MCScanX, accessed on 18 September 2023) and Circos 0.69-9 software were used to analyze the duplication events of *EgCAT* genes. The substitution ratios (ka/ks) (ka/ks > 1, positive selection; ka/ks = 1, neutral selection; ka/ks < 1, negative selection) were calculated to analyze the *EgCATs* duplication. The calculated ka/ks ratio values of *EgCAT1* and *EgCAT3* were 0.241436 (<1) (Appendix A). The results indicated that the evolution of oil palm CAT genes happened via purifying selection and the tandem duplications between chromosomes 4 and 16 during evolution (Figure 5). Finally, the results demonstrated the prominent role of tandem duplication in the expansion of CAT genes in the oil palm genome.

### 2.6. Analysis of cis-Acting Elements in EgCAT Gene Promoter Regions

To understand the presence of *cis*-acting elements, a 2000 bp upstream region of the *EgCAT* genes was downloaded and analyzed via the PlantCARE database. We obtained a series of *cis*-regulatory elements, including ABRE (ABA-response element), TATA-box (MeJA responsive element), and AC-rich (gibberellin responsive element) (Figure 6 and Appendix A), which existed in *EgCAT1-4* genes. These *cis*-acting elements were the key components of hormone-correlated stress responsiveness. The results indicated that *EgCAT* genes may respond to hormone stimulations.

Furthermore, we found ARE (anaerobic responsive element), CAAT-box (abscisic acid-responsive element), and DRE (Dehydration responsive element) in promoter regions of *EgCAT1-4* genes (Figure 6 and Appendix A). In addition, a series of light-responsive elements were identified, indicating the key role of *EgCATs* in response to light stress. Generally, *EgCAT1-4* genes containing hormone/stress-related elements may play a crucial role in resisting hormonal and abiotic stresses.

### 2.7. Prediction of miRNA-Targeting Genes in Oil Palm Genome

MicroRNAs (miRNAs) are a class of non-coding endogenous small molecule RNAs with 19–24 nt. Functional studies have shown that miRNAs are involved in the regulation of all cellular processes, including growth and development, stress response, metabolism, and signal transduction. Recent studies have found that plant miRNAs, as a class of widely functional molecular regulatory factors, regulate the expression of plant genes. Therefore, to clarify the relationship between microRNAs (miRNAs) and *EgCAT* genes, the CDS of the *EgCAT* genes were used to search for putative target sites of miRNAs. As shown in Figure 7, seven putative miRNAs targeting four *EgCAT* genes were identified. We found that *miR396x/y* targeted the *EgCAT1* exon, *EgCAT2* was targeted by *miR159x* and *miR156*, whereas *miR159x* targeted to exon and *miR156* targeted the 3′ UTR region. *miR172a/b* targeted to the exon of *EgCAT3* and *miR156* targeted to the 3′ UTR region of *EgCAT3*. *miR2118-y*, *miR535x*, and *miR168x* targeted to exons of *EgCAT4*, while *miR535x* targeted to the 3′ UTR of *EgCAT4* (Figure 7, Appendix A).

### 2.8. Expression Profile Analysis of EgCAT Genes

To explore the transcript levels of *EgCAT* genes in root (SRR851071), shoot (SRR851103), leaf (SRR851096), flower (SRR851108), fruit (SRR851067), and mesocarp from four developmental stages (15 (SRR190698), 17 (SRR190699), 21 (SRR190701), and 23 (SRR190702) weeks old) based on RNA-seq data from oil palm. Heatmap analysis showed that similar expression patterns for root, shoot, flower vs. leaf, fruit, and mesocarp were noticed for *EgCAT* genes. *EgCAT1* with the highest expression was observed in the leaf (Figure 8, Appendix A). *EgCAT2* showed higher expression in mesocarp tissue of 15-, 17-, 21-, and 23-week-old oil palms. *EgCAT3* had the highest expression level in the mesocarp of 21- and 23-week-old, and *EgCAT4* expressed higher in the mesocarp of 15-week-old. However, *EgCAT5* was not expressed in all tissues of the oil palm. Overall, the result indicated that *EgCAT* family members exhibited mixed tissue expression in oil palms (Figure 8).

### 2.9. Expression Analysis of EgCAT Genes of Oil Palm under Abiotic Stresses

To further identify the responses to abiotic stresses (cold, drought, salt) of *EgCAT* genes we examined the mRNA expression profiles of the five *EgCAT* genes at the molecular level, via qRT-PCR (Figure 9). Notably, except for *EgCAT5*, the *EgCAT1-4* genes showed relatively high expression levels under cold, drought, and salinity stress treatments at a certain time (4 h, 24 h, and 48 h). Under the cold stress condition, *EgCAT1-4* genes showed higher expression patterns at 4 h, 24 h, and 48 h. The drought stress induced the up-regulation of *EgCAT1*, *EgCAT2,* and *EgCAT4*. *EgCAT1-3* genes had a higher expression at 4 h, 24 h, and 48 h under salt stress conditions (Figure 9). Accordingly, the results suggested that *EgCAT* family members may be involved in regulating the response to cold, drought, and salinity stresses in oil palm.

## 3. Discussion

Catalase (CAT) is a crucial enzyme in eukaryotic antioxidant systems, it is primarily found in peroxisomes and plays a vital role in the clearance of reactive oxygen species (ROS) [28]. CAT genes have multiple functions in plants, including resisting adverse environments and protecting plant cells from ROS toxicity [29]. High-throughput sequencing technologies have rapidly advanced, providing insights into the latest research platform for identifying and analyzing CAT family members [30]. Exploring the biological functions of *EgCAT* genes and their abiotic stress regulation mechanisms can aid in the development of new oil palm varieties with enhanced resistance. However, none of the genome-wide studies have explored the structure and role of CAT family genes in oil palm. This is the first report on the identification of CAT transcription factors in oil palm. Moreover, we evaluated their function in response to diverse abiotic stimuli via real-time PCR analysis.

In the current study, five *EgCAT* genes were obtained via genome-wide analysis, which was greater than the number of CAT genes identified in Arabidopsis (3) [31], rice (3) [32], maize (3) [33], cucumber (3) [34], and Hordeum vulgare (2) [16] genomes, but less than that in cotton (7) [31] and wheat (10) [35] genomes. The oil palm CAT gene structural analysis revealed that *EgCAT1-4*, consisting of four *EgCAT* family members, has seven introns, while *EgCAT5* has only three introns. The oil palm CAT gene structure analysis results were consistent with the other CAT structural organizations observed in various plants [25,30]. Analysis of conserved motifs is essential to understanding transcription factor DNA-binding activity, protein–protein interactions, and transcriptional activity [17]. The conserved motif analysis of all *EgCAT* family members revealed the occurrence of five conserved motifs, which may be related to specific functions shared among *EgCAT* family members. The phylogenetic analysis of oil palm CAT family members, including *Arabidopsis thaliana*, *Oryza sativa*, *Olea europaea* subsp. europaea, and *Olea europaea* var. sylvestris showed the conservation and diversification of CAT members. Our research revealed that tandem duplications in the oil palm genome occur for the expansion of all CAT family members during evolution, emphasizing the commonality of gene duplication events in plants [36].

Numerous reports have demonstrated that CAT family members are widely expressed in various tissues/organs for regulating the growth and development of plants [37]. The tissue-specific expression profiling data of genes are vital for elucidating the functional roles of genes. The analysis of heat maps revealed the expression of *EgCAT* members in various parts of oil palm plants. The *EgCAT* family members have demonstrated their tissue-specific expression, indicating their role in regulating growth and developmental processes in oil palm plants. MiRNA (microRNA) is a non-coding small RNA found in organisms with regulatory effects [38,39]. Research on miRNAs in woody plants primarily focuses on their role in regulating biological activities like plant growth, development, signal transduction, stress response, and secondary metabolite generation. [40,41]. There have been reports on the relationship between miRNA and target genes in wheat [25] and cotton [30]. The study shows that eight miRNAs regulate four *EgCAT* genes through potential miRNA-targeting sites in the *EgCAT* genes. These miRNAs include *miR396x/y* targeting *EgCAT1*; *miR159x* and *miR156* targeting *EgCAT2*; *miR172a/b* and *miR156* targeting *EgCAT3*; *miR2118y*, *miR535x*, *miR168x,* and *miR535x* targeting *EgCAT4*. The results suggested that miRNAs may play important roles in regulating the transcript level of CAT genes in oil palm.

Previous studies have shown the regulatory role of CAT members in response to abiotic stress conditions [41,42,43]. In our study, we examined the expressions of five *EgCAT* family members under cold, drought, and salinity stress conditions, revealing all tested transcription factors’ expressions. In this study, we found that cold stress significantly increased the expression of *EgCAT1-4* genes, aligning with previous reports on cold-induced CAT transcription factors [44]. Higher expression of *EgCAT1-2* and *EgCAT4* genes was also observed under drought stress conditions. Our findings are consistent with earlier research that has found drought-induced CAT family members [45]. The present study showed a higher expression of *EgCAT1-3* genes under salinity stress environments. Our results relate to the earlier reports on salt stress-induced CAT family members in various plants [25,41,46]. These findings are helpful in choosing potential CAT family members for producing abiotic stress-tolerant oil palm varieties.

## 4. Materials and Methods

### 4.1. Identification of EgCAT Genes

The genome sequences of oil palm (*Elaeis guineensis*, Jacq.) were obtained from the National Center for Biotechnology Information (NCBI) database. The known CAT amino acid sequences of *Arabidopsis thaliana*, *Oryza sativa* (ssp. Japonica and Indica), *Olea europaea* subsp. europaea, and *Olea europaea* var. sylvestris were downloaded from the Plntfdb (http://plntfdb.bio.uni-potsdam.de/v3.0/, accessed on 13 September 2023) and PlantTFDB (http://planttfdb.gao-lab.org/, accessed on 13 September 2023) databases, and used as the inquiry in the Basic Local Alignment Search Tool (BLAST) program to search for CAT genes in the oil palm genome. The conserved catalase-related immune-responsive domains (PF06628) corresponding to CAT gene family members were obtained from the Pfam database (https://pfam.xfam.org/, accessed on 13 September 2023). The conserved domains database (CDD) online tool (https://www.ncbi.nlm.nih.gov/cdd/, accessed on 14 September 2023) was employed to verify the candidate gene sequences obtained from PFAM. Finally, a total of five *EgCAT* genes were identified in the oil palm genome. The ExPASy proteomic website (https://web.expasy.org/compute_pi/, accessed on 14 September 2023) was employed to predict the molecular weight (MW) and isoelectric points (pI) of oil palm CAT proteins. Subcellular localization of five EgCAT proteins was predicted using the CELLO tool (http://cello.life.nctu.edu.tw/, accessed on 14 September 2023).

### 4.2. EgCAT Gene Structure and Conserved Motif Analysis

The intron–exon organization of *EgCAT* was constructed with TBtools 1.1.0 software (https://github.com/CJ-Chen/TBtools, accessed on 16 September 2023). MEME 5.5.5 online software (https://meme-suite.org/meme/doc/meme.html, accessed on 16 September 2023) was used to identify the conserved motifs of the *EgCATs.*

### 4.3. Phylogenetic Relationship, Chromosomal Distribution, and cis-Acting Elements Prediction of EgCAT Genes

A maximum likelihood (ML) phylogenetic tree of CATs from *Elaeis guineensis*, *Arabidopsis thaliana*, *Oryza sativa*, *Olea europaea* subsp. europaea, and *Olea europaea* var. sylvestris was constructed via the neighbor-joining method, using the MEGA11.0 software based on amino acid sequences of the conserved CAT domain and with 1000 bootstrap replicates for reliability. The chromosomal distribution of CATs was investigated against the oil palm genome using the TBtools software (https://github.com/CJ-Chen/TBtools, accessed on 20 September 2023). The potential cis-regulatory elements of oil palm CAT genes were obtained by choosing 2000 bp upstream of the transcription start site in each CAT gene. The PlantCARE online tool (http://bioinformatics.psb.ugent.be/webtools/plantcare/html/, accessed on 20 September 2023) was employed to predict the cis-acting regulatory elements.

### 4.4. Tissue-Specific Expression of EgCAT Genes Based on Available Transcriptome Datasets

The available transcriptome data of oil palm tissues, including shoot (SRR851103), leaf (SRR851096), root (SRR851071), fruit (SRR851067), flower (SRR851108), and mesocarp (15 weeks (SRR190698), 17 weeks (SRR190699), 21 weeks (SRR851067), and 23 weeks (SRR190702)), were downloaded from the SRA (Sequence Read Archive) database of the NCBI website. The transcript abundance of *EgCATs* in different tissues was calculated with RPKM (reads per kilobase per million mapped values). TBtools 1.1.0 (https://github.com/CJ-Chen/TBtools, accessed on 21 September 2023) was used to generate a heatmap of the *EgCATs*.

### 4.5. Plant Material and Abiotic Stress Treatments

To analyze the expression profile of *EgCAT* genes under cold, drought, and salt stresses, a total of 36 oil palm plantlets were grown in greenhouses (27 °C, 16 h/light, 8 h/darkness, humidity of about 50–60%) at the Coconut Research Institute, Chinese Academy of Tropical Agricultural Sciences, Wenchang, China. Healthy oil palm plantlets of the same age (6 months old) were chosen for cold, salt, and drought treatments. Each abiotic stress treatment (cold, drought, or salinity) was performed after 0 h (control), 4 h, 24 h, and 48 h, using different seedlings (nine for each treatment). For cold stress treatment, oil palm seedlings were exposed to cold stress exposure at 8 °C. Drought stress conditions for the oil palm seedlings were established after reaching 20% water content in the soil. Salinity stress was induced by immersing the roots of oil palm seedlings in 300 mmol/L concentration of NaCl solution. At different time intervals (0, 4, 24, and 48 h) of the above stress exposures, the spear leaves were collected and immediately frozen in liquid nitrogen for further RT-qPCR analysis. In addition, oil palm seedlings for control experiments were maintained under light (16 h) and darkness (8 h) at 27 °C. All the stress experiments used three biological replicates and were repeated three times.

### 4.6. RNA Isolation and Real-Time qPCR Assays

The real-time PCR expression analysis was performed to investigate the relative expression of CAT genes in leaf samples exposed to different abiotic stress conditions. Total RNA was extracted from the leaves collected under normal and abiotic stress conditions using Trizol reagent (Invitrogen, Carlsbad, CA, USA). The RIN values of extracted RNA from normal and stress-exposed samples ranging from 7 to 8 were used for further cDNA synthesis. The isolated RNA (~2 μg) was then reverse transcribed to cDNA using a MightyScript Plus first-strand cDNA synthesis kit (gDNA digester) (Sangon Biotech, Shanghai, China). Quantitative real-time PCR reactions were carried out using a standard SYBR Premix (TaKaRa, Dalian, China) with Mastercycler (Eppendorf, Hamburg, Germany). The oil palm Actin1 gene was used as a housekeeping gene to check the relative expression of SPLs by employing the 2^−ΔΔCt^ method. The information on oil palm CAT gene-specific primers used for qPCR experiments is listed in Appendix A.

### 4.7. Statistical Analysis

Three biological and technical repeats were performed to determine the reliability of the gene expression studies. One-way analysis of variance (ANOVA) was used to determine the statistical significance at the *p *≤** 0.05 and p ≤ 0.01 levels. Asterisks represent a significant difference at *p* ≤ 0.05 (*) and *p* ≤ 0.01 (**).

## 5. Conclusions

To the best of our knowledge, this is the first report on the identification of *EgCAT* transcription factors from the African oil palm genome through bioinformatic analysis. Our bioinformatic approach revealed the CAT genes’ structural organization, their distribution across 16 chromosomes, motif conservation among all the family members, and duplication events among all CAT gene family members. Additionally, we predicted subcellular localization and analyzed the cis-acting regulatory elements in promoter regions and the tissue-specific expression of oil palm CAT family members. In addition, we identified the interaction sites between miRNA and the target *EgCAT* genes. Furthermore, our real-time expression analysis revealed their expression under cold, drought, and salinity conditions. Overall, our bioinformatic, as well as expression analyses, provide valuable information on oil palm CAT genes for extending their role in abiotic stress-tolerant studies of oil palm in the near future.

## Figures and Tables

**Figure 1 ijms-25-01480-f001:**
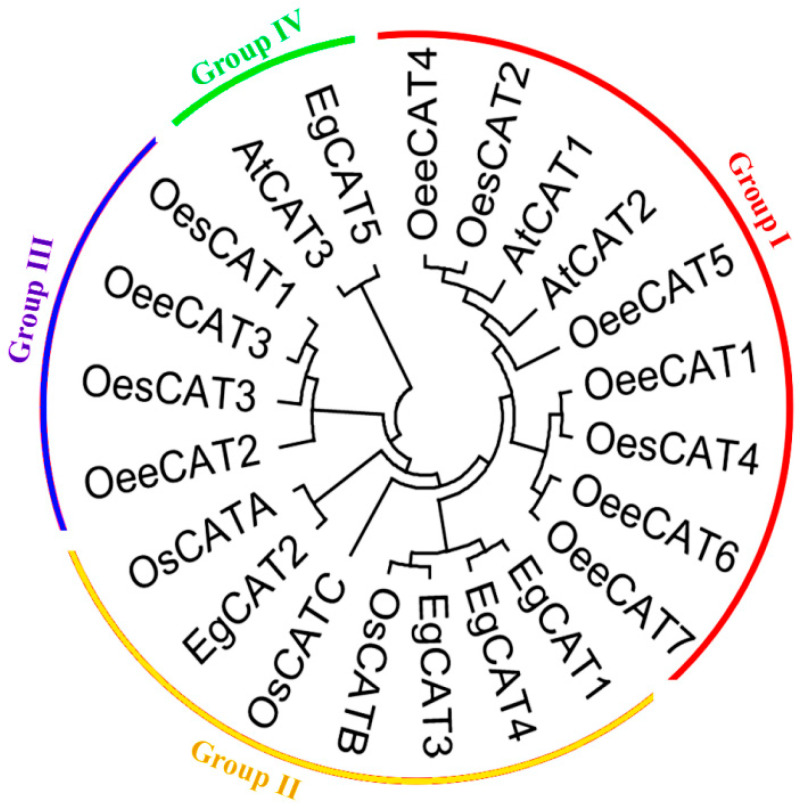
Phylogenetic analysis of CAT genes in oil palm, *Arabidopsis thaliana*, *Oryza sativa*, *Olea europaea* subsp. europaea, and *Olea europaea* var. sylvestris.

**Figure 2 ijms-25-01480-f002:**
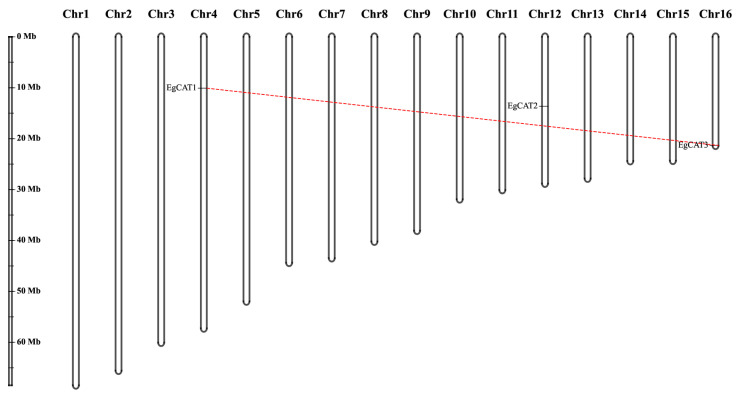
Chromosome mapping of three *EgCAT* genes of oil palm. The length of the chromosome is indicated on the vertical greyscale. The chromosome numbers (1–16) are named and located on the top of each chromosome.

**Figure 3 ijms-25-01480-f003:**
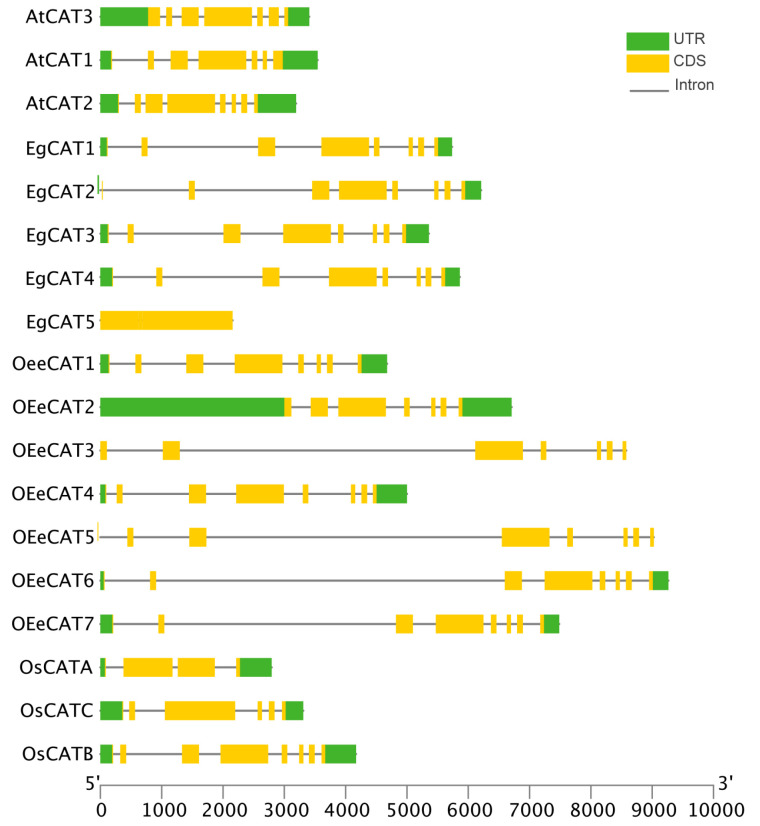
Gene structure analysis of CAT genes of oil palm, *Arabidopsis thaliana*, *Oryza sativa*, *Olea europaea* subsp. europaea and *Olea europaea* var. sylvestris. The blue box represents the upstream/downstream sequences, and the yellow boxed and black horizontal lines represent the exons and introns, respectively.

**Figure 4 ijms-25-01480-f004:**
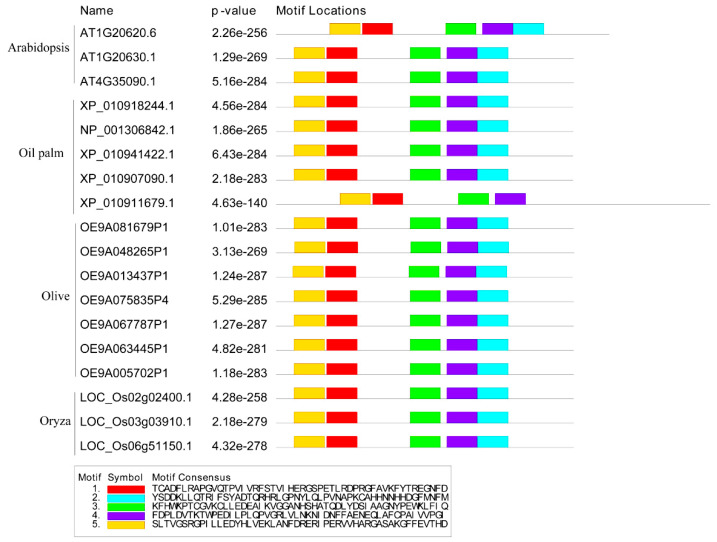
The architecture of the conserved domain of oil palm CAT proteins along with Arabidopsis, olive, and *Oryza sativa*. The abundance of amino acids in each motif of CAT proteins was given in the sequence logo.

**Figure 5 ijms-25-01480-f005:**
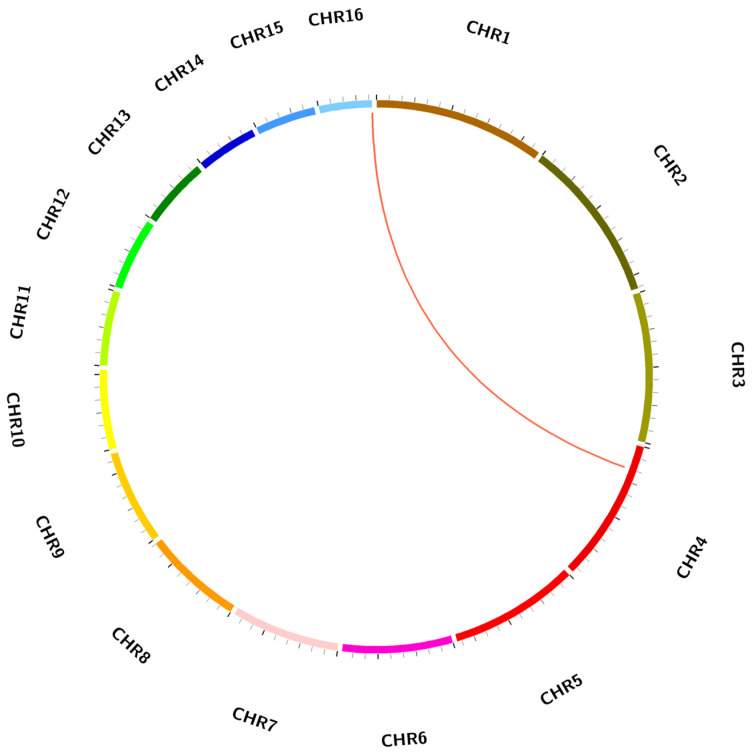
Schematic interpretation of *EgCAT* genes’ duplication in oil palm genome across the chromosomes. Red lines inside the schematic view denote the duplicated gene pairs.

**Figure 6 ijms-25-01480-f006:**
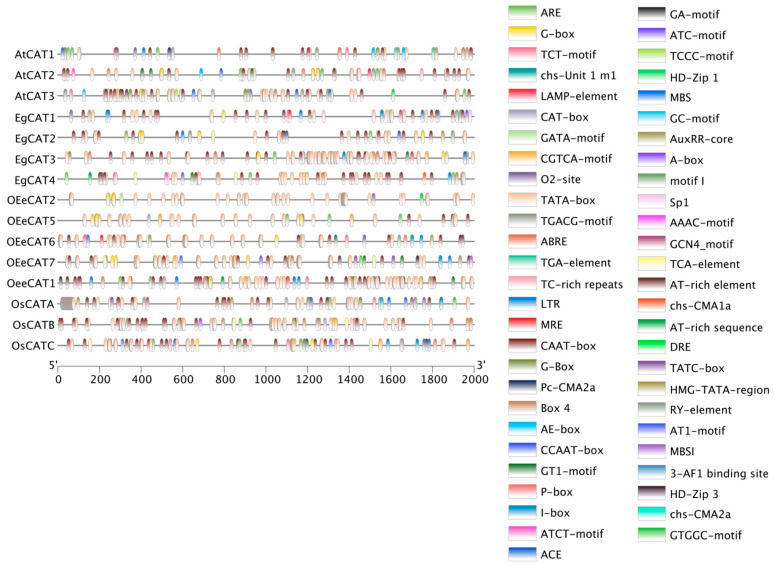
*cis*-acting elements in the promoters of *EgCAT* genes in the oil palm genome. Different color boxes show different identified *cis*-acting elements.

**Figure 7 ijms-25-01480-f007:**
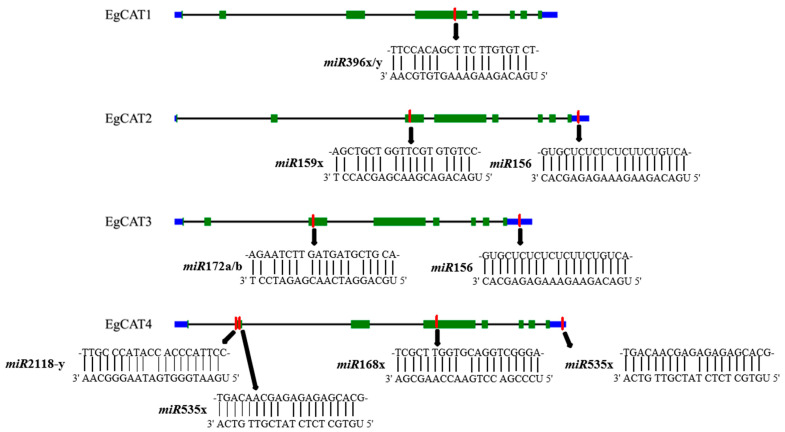
miRNA targeting *EgCAT* genes analysis of oil palm. The blue box represented the UTR regions, and the green box represented the exons. The red boxes were miRNA complementary sites. The black arrow pointed to the specific miRNA. The sequences of miRNA and the complementary *EgCAT* sequences were shown in the expanded regions.

**Figure 8 ijms-25-01480-f008:**
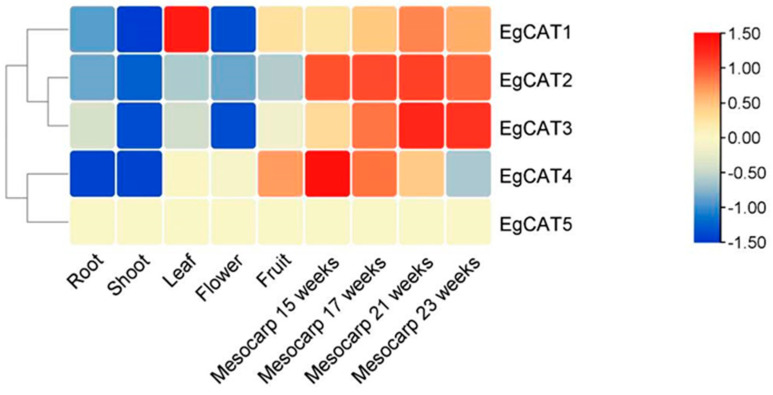
Heatmap for oil palm *CAT* gene expression in tissues, such as root (SRR851071), shoot (SRR851103), leaf (SRR851096), flower (SRR851108), fruit (SRR851067), and mesocarp, from four developmental stages (15 (SRR190698), 17 (SRR190699), 21 (SRR190701), and 23 (SRR190702) weeks old) of oil palm plants The color scale represents the expression levels (low or high) of *EgCAT* genes.

**Figure 9 ijms-25-01480-f009:**
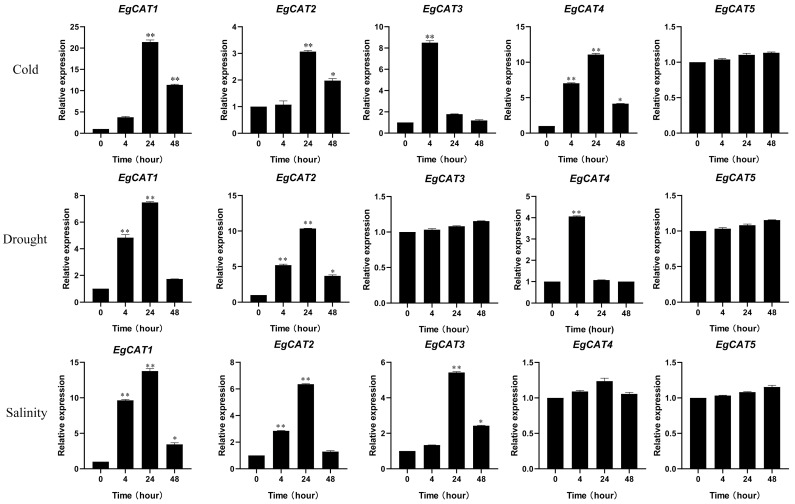
Relative expression of five oil palm *EgCAT* genes under abiotic stress conditions (cold, salt, and drought stress) upon exposures of 0, 4, 24, and 48 h duration, respectively. Asterisks represent significant difference, at *p* ≤ 0.05 (*) and *p* ≤ 0.01 (**).

## Data Availability

The data presented in this study are available on request from the corresponding author.

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
