# Peer review of "Catalase (CAT) Gene Family in Oil Palm (Elaeis guineensis Jacq.): Genome-Wide Identification, Analysis, and Expression Profile in Response to Abiotic Stress"

_ijms, 2024, doi:10.3390/ijms25031480_

Round 1

Reviewer 1 Report

Comments and Suggestions for Authors

The purpose of the presented work was to identify catalase genes and corresponding regulatory elements of oil palm, as well as their role in the response to abiotic stress factors. To do this, the authors analyzed genomic and transcriptomic data obtained previously. As a result, 5 EgCAT genes were discovered, their structure was studied, and phylogenetic analysis was performed. The authors also presented an assessment of their function in response to various abiotic stimuli using real-time PCR analysis. It is further concluded that this is the first report of the identification of CAT transcription factors in oil palm.

In my opinion, the functional data are speculative and require more rigorous experimental confirmation. The authors did not present any protein transcription factors, while the listed non-coding RNAs have different sources and conclusions about their function are based on hypothetical reasoning. The only qRT-PCR experiment whose data are presented suffers from many flaws that raise doubts about the reliability of the results. Appendix 6 shows four pairs of primers, while Figure 9 gives values for 5 genes. In the description of the method there is no mention of normalization of data by reference genes and information about them. The description of temperature cycles includes the stage of primer annealing at 55 degrees, while the calculated annealing temperature of the presented primers exceeds 66 degrees. Careless handling of data can also be noted in relation to the construction of a heat map based on published transcriptomic data. These data were obtained on the 454 instrument in 2013 (Orion Genomics) and have too low coverage for modern requirements. The scale (probably Log2 Fold Changes) range of +/- 1.5 used by the authors does not correspond to the conclusions about “significant differences in the expression patterns of EgCAT genes among different tissues” (line 216).

The relevance of references deserves special note. What does the frequency of allelic polymorphism of the CAT gene in Russian and Buryat adolescents have to do with abiotic stress in plants (line 26, link 1)? What is the relationship between Graves' disease patients and plant growth and development (line 35, reference 5)? What does the data on the regulation of ganodermic acid content in fungi or γ-aminobutyric acid production in lactobacilli confirm regarding the participation of catalase genes in abiotic stress in oil palm (line 84, references 26, 27)? How do CAT single nucleotide polymorphisms in miners correlate with plant resistance to adverse environmental conditions(line 245, reference 32)? These are not all the examples that can be given.

Based on the above, it should be concluded that the manuscript does not contain new valuable information, does not meet the necessary requirements and cannot be considered for publication.

Author Response

Dear editors and anonymous reviewers

We are glad to receive your valuable comments and suggestions to our manuscript. Thank you very much for your kind consideration on this manuscript “Catalase (CAT) gene family in oil palm (Elaeis guineensis Jacq.): Genome-wide identification, analysis and expression profile in response to abiotic stress”. Without your professional reviews, this manuscript would not be as smooth as what it is now. Thank you very much! 

We added an author (Rajesh Yarra) in our manuscript, because he mainly contributed to english editing of the whole manuscript, content modification, and the response of review comments, so hope editors and reviewers agree us to add him as an author. Thank you very much!

This is to confirm that we have amended the manuscript according to all the opinions, suggestions and comments and all the changes have been marked-up in the text by the red colored fonts. The responses to all the comments and suggestions are itemized as follows:

Reviewer 1

Comment 1:The purpose of the presented work was to identify catalase genes and corresponding regulatory elements of oil palm, as well as their role in the response to abiotic stress factors. To do this, the authors analyzed genomic and transcriptomic data obtained previously. As a result, 5 EgCAT genes were discovered, their structure was studied, and phylogenetic analysis was performed. The authors also presented an assessment of their function in response to various abiotic stimuli using real-time PCR analysis. It is further concluded that this is the first report of the identification of CAT transcription factors in oil palm.

Response: Thanks for your kind words.

Comment 2:In my opinion, the functional data are speculative and require more rigorous experimental confirmation. The authors did not present any protein transcription factors, while the listed non-coding RNAs have different sources and conclusions about their function are based on hypothetical reasoning. The only qRT-PCR experiment whose data are presented suffers from many flaws that raise doubts about the reliability of the results. Appendix 6 shows four pairs of primers, while Figure 9 gives values for 5 genes. In the description of the method there is no mention of normalization of data by reference genes and information about them. The description of temperature cycles includes the stage of primer annealing at 55 degrees, while the calculated annealing temperature of the presented primers exceeds 66 degrees. Careless handling of data can also be noted in relation to the construction of a heat map based on published transcriptomic data. These data were obtained on the 454 instrument in 2013 (Orion Genomics) and have too low coverage for modern requirements. The scale (probably Log2 Fold Changes) range of +/- 1.5 used by the authors does not correspond to the conclusions about “significant differences in the expression patterns of EgCAT genes among different tissues” (line 216).

Response: Thanks for your useful and professional comments. We only concentrated on bioinformatics analysis and hyptothetical predictions based on bioinformatic study. Currently, we are working on functional studies and will report it in separate paper in future. Sorry for the mistakes regarding qPCR study. Now we add the primer 5 pair. We have revised the total qPCR methodology. There is a mistake on writing about methodology regarding qPCR experiments. Ye, we agree with the transcriptome data obtained in 2013, but still its reliable. The way of our writing expressions leads to negative conclusions. We corrected it as suggested.

Comment 3:The relevance of references deserves special note. What does the frequency of allelic polymorphism of the CAT gene in Russian and Buryat adolescents have to do with abiotic stress in plants (line 26, link 1)? What is the relationship between Graves' disease patients and plant growth and development (line 35, reference 5)? What does the data on the regulation of ganodermic acid content in fungi or γ-aminobutyric acid production in lactobacilli confirm regarding the participation of catalase genes in abiotic stress in oil palm (line 84, references 26, 27)? How do CAT single nucleotide polymorphisms in miners correlate with plant resistance to adverse environmental conditions (line 245, reference 32)? These are not all the examples that can be given.

Response: Thanks for clear observation. Sorry for citing those irrelevant references. Now we removed those references and added relevant references as suggested.

We have amended the manuscript according to all your suggestions. Thanks again for your quick processing and professional editing of this manuscript. What you have done will always be highly and greatly appreciated. Any questions, we will be more than happy to answer. Looking forward to hearing from you soon. Best wishes!

  Coconut Research Institute, Chinese Academy of Tropical Agricultural Science, Hainan China

                                                   Jianqiu Ye and Hongxing Cao

                                                           2023-12-29

Reviewer 2 Report

Comments and Suggestions for Authors

Zhou et al conducted genome wide characterization of Catalase (CAT) gene family in oil palm (Elaeis guineensis Jacq.). The manuscript provides information. However, there are few crucial points that if considered will increase the accuracy, value of the manuscript and may be readability.

-Please add the future prospect of the work, if there is any to the abstract and introduction.

-Please mention the abiotic stress used in the study in the abstract.

-Please provide information on the quality of isolated RNAs. For example, their gel picture or RIN values. You can cite proper studies while explaining the quality of the isolated RNA. For example, https://www.mdpi.com/2073-4395/13/3/631 or https://www.mdpi.com/2073-4395/12/10/2421. RNA quality is important for such studies. Any contamination may affect the reliability of the obtained results.

-In conclusion, authors mentioned ‘Furthermore, our experimental approach validated their role under various abiotic stress conditions.’ Please explain how. Authors did not conduct the functional analysis.

-Tables and Figure captions should be self-explanatory. Please elaborate them by mentioning the experimental plant name, treatments etc. Applicable for all the captions.

I do believe that the manuscript can be accepted once the authors address the mentioned points and enrich the manuscript with the crucial information. 

Comments on the Quality of English Language

Moderate editing of English language required

Author Response

Dear editors and anonymous reviewers

We are glad to receive your valuable comments and suggestions to our manuscript. Thank you very much for your kind consideration on this manuscript “Catalase (CAT) gene family in oil palm (Elaeis guineensis Jacq.): Genome-wide identification, analysis and expression profile in response to abiotic stress”. Without your professional reviews, this manuscript would not be as smooth as what it is now. Thank you very much! 

We added an author (Rajesh Yarra) in our manuscript, because he mainly contributed to english editing of the whole manuscript, content modification, and the response of review comments, so hope editors and reviewers agree us to add him as an author. Thank you very much!

This is to confirm that we have amended the manuscript according to all the opinions, suggestions and comments and all the changes have been marked-up in the text by the red colored fonts. The responses to all the comments and suggestions are itemized as follows:

Reviewer 2

Comment 1:Zhou et al conducted genome wide characterization of Catalase (CAT) gene family in oil palm (Elaeis guineensis Jacq.). The manuscript provides information. However, there are few crucial points that if considered will increase the accuracy, value of the manuscript and may be readability.

Response: Thanks for your kind words. Following we will revise it accordingly.

Comment 2:Please add the future prospect of the work, if there is any to the abstract and introduction.

Response: We provided it in abstract now as suggested.

Comment 3:Please mention the abiotic stress used in the study in the abstract.

Response: We provided it in abstract now as suggested.

Comment 4:Please provide information on the quality of isolated RNAs. For example, their gel picture or RIN values. You can cite proper studies while explaining the quality of the isolated RNA. For  example,  https://www.mdpi.com/2073-4395/13/3/631 or https://www.mdpi.com/2073-4395/12/10/2421. RNA quality is important for such studies. Any contamination may affect the reliability of the obtained results.

Response: We provided the information as suggested.

Comment 5:In conclusion, authors mentioned ‘Furthermore, our experimental approach validated their role under various abiotic stress conditions.’ Please explain how. Authors did not conduct the functional analysis.

Response: We corrected the information as suggested.

Comment 6:Tables and Figure captions should be self-explanatory. Please elaborate them by mentioning the experimental plant name, treatments etc. Applicable for all the captions.

Response: We provided the information as suggested.

Comment 7:I do believe that the manuscript can be accepted once the authors address the mentioned points and enrich the manuscript with the crucial information. 

Response: Thanks again and this is to confirm that we have provided all those information suggested.

We have amended the manuscript according to all your suggestions. Thanks again for your quick processing and professional editing of this manuscript. What you have done will always be highly and greatly appreciated. Any questions, we will be more than happy to answer. Looking forward to hearing from you soon. Best wishes!

  Coconut Research Institute, Chinese Academy of Tropical Agricultural Science, Hainan China

                                                   Jianqiu Ye and Hongxing Cao

                                                           2023-12-29

Reviewer 3 Report

Comments and Suggestions for Authors

The manuscript titled “Catalase (CAT) gene family in oil palm (Elaeis guineensis Jacq.): Genome-wide identification, analysis and expression profile in response to abiotic stress” reports a study on  the identification and response to abiotic stress of CAT family genes. This study is important and interesting, whereas these were existed several deficiencies as below:

1 In the abstract, the abbreviation of ROS is unnecessary as it only occurred once..

2 CAT in oil palm should be EgCAT. Please unify the whole text.

3 Authors analyzed the expression of EgCATs in leaves responded to abiotic stress. Why choose leaves for verification?

4 In figure 2, all five CATs should be displayed on the chromosomes.

5 Oil palm should be noted for the first time

6 The font size in figure 3 were too small, they should be adjusted to provide adequate legibility.

7 What is the difference between Figure 1 and 5? In figure 5, all genes duplication in oil palm genome should be displayed.

8 Latin Name of some specials were mistake: tomato should be Solanum lycopersicum L., tobacco should be Nicotiana tabacum L..

9 Line 48, Brassica napus should be italic.

10 In section 2.2, authors show “Significantly, EgCAT genes had a closer phylogenetic relationship with OsCAT and AtCAT.” It is unclear how  this conclusion was reached?

11 The legend of figure 3 was incomplete. The meaning of the black line should be marked.

12 RT-qPCR should be conducted to verify the results presented in Figure 8.

13 Enzyme activity test of the CAT enzyme should be performed.

14 The legend of Figure 9 was too brief. The statistical analysis and the sample methods must be clearly labeled..

15 Line 65, CAT2 should be cat2.

16 line 75, a space should be added between ‘CsCAT1 and CsCAT2’.

17 line 52, SPCAT1 should not italic.

15 Significantly, EgCAT genes had a closer phylogenetic relationship with OsCAT and AtCAT.

Comments on the Quality of English Language

A professional editing service shall be employed to revise the language.

Author Response

Dear editors and anonymous reviewers

We are glad to receive your valuable comments and suggestions to our manuscript. Thank you very much for your kind consideration on this manuscript “Catalase (CAT) gene family in oil palm (Elaeis guineensis Jacq.): Genome-wide identification, analysis and expression profile in response to abiotic stress”. Without your professional reviews, this manuscript would not be as smooth as what it is now. Thank you very much! 

We added an author (Rajesh Yarra) in our manuscript, because he mainly contributed to english editing of the whole manuscript, content modification, and the response of review comments, so hope editors and reviewers agree us to add him as an author. Thank you very much!

This is to confirm that we have amended the manuscript according to all the opinions, suggestions and comments and all the changes have been marked-up in the text by the red colored fonts. The responses to all the comments and suggestions are itemized as follows:

Reviewer 3

Comment 1:The manuscript titled “Catalase (CAT) gene family in oil palm (Elaeis guineensis Jacq.): Genome-wide identification, analysis and expression profile in response to abiotic stress” reports a study on  the identification and response to abiotic stress of CAT family genes. This study is important and interesting, whereas these were existed several deficiencies as below.

Response: Thanks for your kind words. Following we will revise it accordingly.

Comment 2: In the abstract, the abbreviation of ROS is unnecessary as it only occurred once.

Response: We corrected it as suggested.

Comment 3: CAT in oil palm should be EgCAT. Please unify the whole text.

Response: We corrected it as suggested.

Comment 4: Authors analyzed the expression of EgCATs in leaves responded to abiotic stress. Why choose leaves for verification?

Response: Thanks for this comment. Accelerated senescence and leaf abscission are primary symptoms in response to stress and it is easy to isolate RNA from leaves rather than other tissues in oil plam. So we choose leaves as experimental samples for verification.

Comment 5: In figure 2, all five CAT’s should be displayed on the chromosomes.

Response: Thanks for this comment. EgCAT4 and EgCAT5 were distributed on two scaffold regions (p5_sc00387 and p5_sc33453) and have not yet been incorporated onto the physical map of chromosomes.

Comment 6:Oil palm should be noted for the first time

Response: We corrected it as suggested.

Comment 7:The font size in figure 3 were too small, they should be adjusted to provide adequate legibility.

Response: We tried our best to correct it as suggested.

Comment 8: What is the difference between Figure 1 and 5? In figure 5, all genes duplication in oil palm genome should be displayed.

Response: Thanks for this comment. Yes, fig.1 depicts the phylogentic relationship between EgCATs with other plant species, whereas fig.5 depicts the duplication of EgCATs across the oil palm chromosomes.

Comment 9:Latin Name of some specials were mistake: tomato should be Solanum lycopersicum L., tobacco should be Nicotiana tabacum L..

Response: Thanks for this comment. We corrected the tomato but for tobacco the species is Nicotiana plumbaginifolia, not the tabacum species. Thanks for the comments

Comment 10:Line 48, Brassica napus should be italic.

Response: We corrected it to be italic.

Comment 11:In section 2.2, authors show “Significantly, EgCAT genes had a closer phylogenetic relationship with OsCAT and AtCAT.” It is unclear how this conclusion was reached?

Response: Thanks for this comment. We rectified it and modified the language expression.

Comment 12: The legend of figure 3 was incomplete. The meaning of the black line should be marked.

Response: We corrected figure 3 as suggested.

Comment 13: RT-qPCR should be conducted to verify the results presented in Figure 8.

Response: Thanks for this comment. The results presented in fig.8 is based on already available transcriptome data. We performed the expression analysis of identified EgCATs under abiotic stress conditions.

Comment 14:Enzyme activity test of the CAT enzyme should be performed.

Response: Thanks for this comment. Our work is purely related to identification of EgCATs through bioinformatic approach. We are elucidated their role under abiotic stress conditions via qPCR approach.

Comment 15: The legend of Figure 9 was too brief. The statistical analysis and the sample methods must be clearly labeled. 

Response: Thanks for this comment. We corrected it as suggested. Besides, this is to confirm that Figure 9. Relative expression of 5 EgCAT genes under abiotic stress conditions (cold, salt, and drought stress) at 0, 4, 24 and 48 hours, respectively. Asterisks represent significant difference at P ≤ 0.05(*) and P ≤ 0.01(**).

Comment 16:Line 65, CAT2 should be cat2.

Response: We corrected it as suggested.

Comment 17: line 75, a space should be added between ‘CsCAT1 and CsCAT2’.

Response: We corrected it as suggested.

Comment 18: line 52, SPCAT1 should not italic.

Response: We corrected it as suggested.

We have amended the manuscript according to all your suggestions. Thanks again for your quick processing and professional editing of this manuscript. What you have done will always be highly and greatly appreciated. Any questions, we will be more than happy to answer. Looking forward to hearing from you soon. Best wishes!

  Coconut Research Institute, Chinese Academy of Tropical Agricultural Science, Hainan China

                                                   Jianqiu Ye and Hongxing Cao

                                                           2023-12-29

Reviewer 4 Report

Comments and Suggestions for Authors

Interesting work about in silico analysis of a specific gene in oil palm and expression analysis in relation to the cold, salt and drougth stress.

However several importan questions must be revised:

Objectives of the work (lines 83-100) must be summarized and clarified.

Quality of Figure 2, 3, 5, 6 8 and 9 must be improved increasing font size.

Plant material must be btter explained including oil plant genotype and main characteristics.

Experimental design must be clarified indicaniong the deseing of the aboitic stress assays.

qPCR protol must be clarified inlcuing refence genes and main statistical nalayis.

Concñlusions of the work must be improved including main implications of teh obtained results from a breeding and production point of view.

Comments on the Quality of English Language

Minor changes are required

Author Response

Dear editors and anonymous reviewers

We are glad to receive your valuable comments and suggestions to our manuscript. Thank you very much for your kind consideration on this manuscript “Catalase (CAT) gene family in oil palm (Elaeis guineensis Jacq.): Genome-wide identification, analysis and expression profile in response to abiotic stress”. Without your professional reviews, this manuscript would not be as smooth as what it is now. Thank you very much! 

We added an author (Rajesh Yarra) in our manuscript, because he mainly contributed to english editing of the whole manuscript, content modification, and the response of review comments, so hope editors and reviewers agree us to add him as an author. Thank you very much!

This is to confirm that we have amended the manuscript according to all the opinions, suggestions and comments and all the changes have been marked-up in the text by the red colored fonts. The responses to all the comments and suggestions are itemized as follows:

Reviewer 4

Comment 1:Interesting work about in silico analysis of a specific gene in oil palm and expression analysis in relation to the cold, salt and drought stress. However, several important questions must be revised:

Response: Thanks for your kind words. Following we will revise it accordingly.

Comment 2:Objectives of the work (lines 83-100) must be summarized and clarified.

Response: We corrected it as suggested.

Comment 3:Quality of Figure 2, 3, 5, 6 8 and 9 must be improved increasing font size.

Response: We improved it as suggested.

Comment 4:Plant material must be btter explained including oil plant genotype and main characteristics.

Response: We provided the information as suggested.

Comment 5:Experimental design must be clarified indicating the deseing of the aboitic stress assays.

Response: We have already provided it in methods section 4.5

Comment 6:qPCR protol must be clarified inlcuing refence genes and main statistical nalayis.

Response: We provided the information as suggested.

Comment 7:Concñlusions of the work must be improved including main implications of the obtained results from a breeding and production point of view.

Response: Thans for this comment. This is to confirm that in the revised manuscript, we have tried our best to improve the conclusions.

We have amended the manuscript according to all your suggestions. Thanks again for your quick processing and professional editing of this manuscript. What you have done will always be highly and greatly appreciated. Any questions, we will be more than happy to answer. Looking forward to hearing from you soon. Best wishes!

  Coconut Research Institute, Chinese Academy of Tropical Agricultural Science, Hainan China

                                                   Jianqiu Ye and Hongxing Cao

                                                           2023-12-29

Reviewer 5 Report

Comments and Suggestions for Authors

My main objection to the article is the insufficient discussion, which should be greatly expanded. 

Have CAT genes been analyzed in other palm species? What are the differences?

Please compare the evolution of CAT genes among monocotyledons. What are the differences between these genes?

Is the structure of these genes in plants related to palms known?

Why is the paper "Correlation analysis of cold-related gene expression with physiological and biochemical indicators under cold stress in oil palm" (doi: 10.1371/journal.pone.0225768) not cited and discussed?

The authors decently edited the article for correct use of species names and varieties and subspecies. Please check the entire text and correct it. Please add proper author names for latin name of species.

E.g. Olea  europaea L.(subsp. Europaea) Olea europaea var. Sylvestris.

= Olea europaea subsp. europaea  Olea europaea var. sylvestris.

Author Response

Dear editors and anonymous reviewers

We are glad to receive your valuable comments and suggestions to our manuscript. Thank you very much for your kind consideration on this manuscript “Catalase (CAT) gene family in oil palm (Elaeis guineensis Jacq.): Genome-wide identification, analysis and expression profile in response to abiotic stress”. Without your professional reviews, this manuscript would not be as smooth as what it is now. Thank you very much! 

We added an author (Rajesh Yarra) in our manuscript, because he mainly contributed to english editing of the whole manuscript, content modification, and the response of review comments, so hope editors and reviewers agree us to add him as an author. Thank you very much!

This is to confirm that we have amended the manuscript according to all the opinions, suggestions and comments and all the changes have been marked-up in the text by the red colored fonts. The responses to all the comments and suggestions are itemized as follows:

Reviewer 5

Comment 1:My main objection to the article is the insufficient discussion, which should be greatly expanded. 

Response: Thanks for your kind words. Following we will revise it accordingly.

Comment 2:Have CAT genes been analyzed in other palm species? What are the differences?

Response: Thanks. This is to confirm that we did not find any reports regarding the identification of CAT genes in known plam species till date.

Comment 3:Please compare the evolution of CAT genes among monocotyledons. What are the differences between these genes?

Response: Thanks. We have already provided the phylogenetic information between oil palm CAT and the monoctyledon plant rice. It was shown in 2.2 results section and in figure.1.

Comment 4:Is the structure of these genes in plants related to palms known?

Response: Thanks for this comment. We did not have any information regarding CAT genes in known plam species till date. So we are unclear about this.

Comment 5:Why is the paper "Correlation analysis of cold-related gene expression with physiological and biochemical indicators under cold stress in oil palm" (doi: 10.1371/journal.pone.0225768) not cited and discussed?

Response: Thanks for this suggestion. In the revised manuscript, we have cited this paper and briefly discussed it.

Comment 6:The authors decently edited the article for correct use of species names and varieties and subspecies. Please check the entire text and correct it. Please add proper author names for latin name of species.

E.g. Olea europaea L.(subsp. Europaea) Olea europaea var. Sylvestris.

= Olea europaea subsp. europaea Olea europaea var. sylvestris.

Response: Thanks. We edited and corrected the species names and varieties and subspecies as suggested.

We have amended the manuscript according to all your suggestions. Thanks again for your quick processing and professional editing of this manuscript. What you have done will always be highly and greatly appreciated. Any questions, we will be more than happy to answer. Looking forward to hearing from you soon. Best wishes!

  Coconut Research Institute, Chinese Academy of Tropical Agricultural Science, Hainan China

                                                   Jianqiu Ye and Hongxing Cao

                                                           2023-12-29

Round 2

Reviewer 2 Report

Comments and Suggestions for Authors

All the comments have not been properly addressed. 

Comments on the Quality of English Language

Moderate editing of English language required

Author Response

Point-by-point response letter

 Dear editors and anonymous reviewers

We are glad to receive your valuable comments and suggestions to our manuscript. Thank you very much for your kind consideration on this manuscript “Catalase (CAT) gene family in oil palm (Elaeis guineensis Jacq.): Genome-wide identification, analysis and expression profile in response to abiotic stress”. Without your professional reviews, this manuscript would not be as smooth as what it is now. Thank you very much!

This is to confirm that we have amended the manuscript according to all the opinions, suggestions and comments and all the changes have been marked-up in the text by the red colored fonts. The responses to all the comments and suggestions are itemized as follows:

Reviewer 2

Comment 1:All the comments have not been properly addressed.

Response: Now we have addressed as much as to the best of our knowledge

Comment 2:Moderate editing of English language required

Response: We have edited the English language now.

Reviewer 4 Report

Comments and Suggestions for Authors

Authors have revised correctly the manuscript 

Author Response

Point-by-point response letter

 Dear editors and anonymous reviewers

We are glad to receive your valuable comments and suggestions to our manuscript. Thank you very much for your kind consideration on this manuscript “Catalase (CAT) gene family in oil palm (Elaeis guineensis Jacq.): Genome-wide identification, analysis and expression profile in response to abiotic stress”. Without your professional reviews, this manuscript would not be as smooth as what it is now. Thank you very much!

This is to confirm that we have amended the manuscript according to all the opinions, suggestions and comments and all the changes have been marked-up in the text by the red colored fonts. The responses to all the comments and suggestions are itemized as follows:

Reviewer 4

Comment 1:  Authors have revised correctly the manuscript

Response: Thanks for the reviewer comments.

Reviewer 5 Report

Comments and Suggestions for Authors

Unfortunately, I have ethical objections to the behavior of authors, you should not add an additional author for having corrected the English or responded to reviews. I believe that this is not a sufficient substantive contribution. Please remove this additional author!

Author Response

Point-by-point response letter

 Dear editors and anonymous reviewers

We are glad to receive your valuable comments and suggestions to our manuscript. Thank you very much for your kind consideration on this manuscript “Catalase (CAT) gene family in oil palm (Elaeis guineensis Jacq.): Genome-wide identification, analysis and expression profile in response to abiotic stress”. Without your professional reviews, this manuscript would not be as smooth as what it is now. Thank you very much!

This is to confirm that we have amended the manuscript according to all the opinions, suggestions and comments and all the changes have been marked-up in the text by the red colored fonts. The responses to all the comments and suggestions are itemized as follows:

Reviewer 5

Comment 1: Unfortunately, I have ethical objections to the behavior of authors, you should not add an additional author for having corrected the English or responded to reviews. I believe that this is not a sufficient substantive contribution. Please remove this additional author!

Response: We have removed the additional author as suggested, and expressed our thanks in the Acknowledgement section.

Round 3

Reviewer 2 Report

Comments and Suggestions for Authors

-

Comments on the Quality of English Language

Moderate editing of English language required

Reviewer 5 Report

Comments and Suggestions for Authors

After the changes that the authors made to the manuscript, it can be accepted.